# A Peri-Implant Disease Risk Score for Patients with Dental Implants: Validation and the Influence of the Interval between Maintenance Appointments

**DOI:** 10.3390/jcm8020252

**Published:** 2019-02-17

**Authors:** Miguel de Araújo Nobre, Francisco Salvado, Paulo Nogueira, Evangelista Rocha, Peter Ilg, Paulo Maló

**Affiliations:** 1University Clinic of Stomatology, Faculty of Medicine, University of Lisbon, 1649-028 Lisbon, Portugal; fjsalvado2002@yahoo.com; 2Research and Development Department, Maló Clinic, 1600-042 Lisbon, Portugal; 3Institute of Preventive Medicine, Faculty of Medicine, University of Lisbon, 1649-028 Lisboa, Portugal; pnogueira16@gmail.com (P.N.); evangelistarocha@hotmail.com (E.R.); 4Oromaxillofacial Surgery, University of Campinas, São Paulo 13083-970, Brasil; jpilg@uol.com.br; 5Implantology Department, Maló Clinic, 1600-042 Lisbon, Portugal; research@maloclinics.com

**Keywords:** dental implants, peri-implantitis, peri-implant disease, risk, epidemiology

## Abstract

Background: There is a need for tools that provide prediction of peri-implant disease. The purpose of this study was to validate a risk score for peri-implant disease and to assess the influence of the recall regimen in disease incidence based on a five-year retrospective cohort. Methods: Three hundred and fifty-three patients with 1238 implants were observed. A risk score was calculated from eight predictors and risk groups were established. Relative risk (RR) was estimated using logistic regression, and the c-statistic was calculated. The effect/impact of the recall regimen (≤ six months; > six months) on the incidence of peri-implant disease was evaluated for a subset of cases and matched controls. The RR and the proportional attributable risk (PAR) were estimated. Results: At baseline, patients fell into the following risk profiles: low-risk (*n* = 102, 28.9%), moderate-risk (*n* = 68, 19.3%), high-risk (*n* = 77, 21.8%), and very high-risk (*n* = 106, 30%). The incidence of peri-implant disease over five years was 24.1% (*n* = 85 patients). The RR for the risk groups was 5.52 (c-statistic = 0.858). The RR for a longer recall regimen was 1.06, corresponding to a PAR of 5.87%. Conclusions: The risk score for estimating peri-implant disease was validated and showed very good performance. Maintenance appointments of < six months or > six months did not influence the incidence of peri-implant disease when considering the matching of cases and controls by risk profile.

## 1. Introduction

Peri-implant disease is considered a pathological condition occurring in tissues around dental implants, characterized by inflammation in the peri-implant connective tissue and progressive loss of supporting bone [1,2]. The weighted mean prevalence (95% confidence interval) of peri-implant disease is estimated to be 19.83% (15.38, 24.27) at the patient level and 9.25% (7.57, 10.93) at the implant level [3].

Recently, a large case-control study was conducted that examined several risk factors for peri-implant disease: history of periodontitis, bone level, lack of passive fit or non-optimal screw joint, bacterial plaque, bleeding, type of material used in the restoration, proximity of other implants/teeth and current smoking [4]. The same study estimated a potential risk score for predicting peri-implant disease. However, the risk score was not validated in a different cohort than that from where the data were originally collected. 

Profiling patients according to their risk for a disease is currently a priority in public health research, with significant efforts being made to introduce prevention and treatment strategies that account for individual variability [5]. In other medical specialties, profiling a patient’s individual risk for a particular health-related outcome through a risk score has been previously reported, such as the prediction of serious bleeding after primary percutaneous coronary interventions [6], risk scores for predicting adverse reaction to contrast agents for computed tomography [7], or estimation of coronary heart disease risk [8]. 

In Dentistry, one prevention strategy in both Periodontal and Implant Dentistry is the recall regimen, accounting for patients’ signs and symptoms along with patient compliance and acceptance. Recall regimens are designed to help predict the long-term success of an outcome [9,10,11,12,13]. Specifically, a recall interval of five to six months has been proposed as the minimum required for successful outcomes in implant dentistry [12,13,14]. The strength of this recommendation is based on a “D” classification—the lowest classification based on extrapolation from other categories of evidence, evidence from expert committee reports or opinions, clinical experience of respected authorities, or a combination of these categories [14]. Consequently, more research is necessary to produce evidence for recommending recall regimens.

The primary aim of this study was to validate a risk score for peri-implant disease. The secondary aim of this study was to investigate the influence of recall regimens on the incidence of peri-implant disease.

## 2. Materials and Methods

### 2.1. Study Design, Study Population, Ethics, Inclusion and Exclusion Criteria

This retrospective cohort study was approved by the Ethical Committee for Health (authorization 010/2013), the Faculty of Medicine, University of Lisbon Ethical Board (Process 270/2015), and the Faculty of Medicine-University of Lisbon Scientific Board (Process CC-120), and follows the Helsinki Declaration of 1975, as revised in 2000. The patients provided their informed consent to participate in the study. The study was conducted at a private rehabilitation center (Lisbon, Portugal) concerning patient rehabilitation and follow-up and at the Faculty of Medicine, University of Lisbon, between March 2015 and August 2017 (data collection, analysis and interpretation). 

The study population consisted of male and female patients over 18 years of age who received a fixed prosthesis supported by dental implants at a private practice (Lisbon, Portugal) between January and December 2006. The inclusion criterion was patients with rehabilitated dental implants with at least 5 years of follow-up after the insertion of the rehabilitation, healthy or with a systemic condition and irrespective of the smoking status. Exclusion criteria were patients not followed by our team and patients with implants that failed (early complete implant failures during osseointegration, during the first year of follow-up) [15] or had unresolved biological complications (abscess, infection, or fistulae) during the first year of follow-up whose signs and symptoms could overlap with the clinical manifestations of peri-implant disease.

A total of 504 surgery records referring to 479 patients with 1676 implants were reviewed corresponding to the total number of patients rehabilitated between January and December of 2006 at the private practice. Sixty patients with a total of 206 implants were excluded for the following reasons: 43 patients (126 implants) were not followed by our team; 12 patients (60 implants) presented unresolved biological complications during the first year of follow-up; five patients (11 implants) had failures during the first year of follow-up (in addition to 7 implants from two of these patients who were considered survivals but were still not included in the analysis). Additionally, 12 patients (22 implants) had two different surgical interventions for implant rehabilitation in the same year, but with more than 6 months between them. For these patients, only the implants from the first surgical intervention were included in this analysis. Sixty-six patients (18.7%) with 210 implants (17.0%) were lost to follow-up.

### 2.2. Sample Size

The final sample included a total of 353 patients (211 female, 142 male) with an average age of 52.5 years (standard deviation: 11.75 years; age range: 18 to 87 years) and 1238 implants (Brånemark system, NobelSpeedy, Nobel Biocare AB, Göteborg, Sweden). The sample size allowed for the detection of a true relative risk of 1.74 in exposed subjects relative to unexposed subjects, assuming the probability of exposure among controls of 21% [16] with 80% power and a type I error probability of 5% (power and sample size calculations, version 3.0.34, Dupont WD and Plummer WD Jr, Department of Biostatistics, Vanderbilt University, Nashville, TN, USA). 

### 2.3. Outcome Measures, Variables

The primary outcome measure was the model’s discrimination capacity for predicting peri-implant disease. Peri-implant disease (dependent variable) was defined as the presence of peri-implant pockets ≥5 mm, bleeding upon probing, the concurrent presence of vertical bone loss visible in the periapical radiograph compared to the previous evaluation and/or attachment loss ≥2 mm compared to the previous evaluation [4]. The current risk score derived from the construction of a model referred to in a previous publication [16]. In brief, the following steps were taken in the construction of the model: First, a univariable statistical evaluation was performed between potential risk factors and peri-implant disease. The following variables were evaluated considering the potential association with peri-implant disease: 

Classification of comorbidities according the American Society of Anesthesiology (ASA) (ASA I-III); history of head and neck irradiation <6 months; history of chemotherapy <6 months; smoking habits (non-smoker, smoker); history of periodontitis (periodontitis previous to the implant rehabilitation or tooth loss due to periodontitis); postmenopausal hormone replacement therapy; Implant location (mandible, maxilla), Implant position within the jaw (anterior, posterior); implant orientation (axial, tilted), proximity of the implant to teeth or other implants (absence, presence of a natural tooth or implant immediately next to the implant); implant length (7, 8.5, 10, 11.5, 13, 15 or 18 mm); implant diameter (3.3–3.5 mm, 3.75–4.3 mm, 5–6 mm); implant surface (machined, oxidized); cantilevers (absence, 1 or more); implant-crown ratio (2:1, 1:1); type of abutment (straight, angulated 17 degrees, angulated 30 degrees); abutment height (1–5 mm); type of prosthetic restoration (single tooth, fixed partial restoration or fixed total restoration); type of material used in the restoration (ceramic, metal–ceramic, metal–acrylic, acrylic); fracture of prosthetic components within the previous year of diagnosis, lack of prosthetic fit or non-optimal screw joint (defined as the loosening of prosthetic screw or the visible gap between prosthetic components); implants in grafted bone; implants inserted in post-extraction sockets; implants inserted in periodontally compromised sites; phasing of implant restoration (2-stage: implant insertion in a first surgery, followed by a healing period between 4–6 months, and a second surgery for abutment connection; 1-stage: implant and abutment insertion on the same surgery; immediate function: implant, abutment and connection of the restoration on the same day of surgery); presence of bacterial plaque, presence of bleeding, bone level (located on the implants’ coronal, or medium thirds); time in months between clinical evaluation appointments. The variables individually associated with the incidence of peri-implant disease were tested in a multivariable conditional logistic regression model. The variables were introduced following a criterion of biological plausibility (first) and statistical performance (second). The statistical performance was evaluated according to the variables’ statistical significance (*p* < 0.05) and the models’ goodness of fit (Likelihood ratio test). The exceptions were the variables “bacterial plaque” and “smoking habits” that were always included in the model given their potential confounder effect, and this way, the effect of confounding variables was considered to have been reduced by multivariable analysis. Interaction between variables was performed based on biological plausibility and previous scientific reports (bacterial plaque and proximity of the implant to other teeth or implants). The final model was derived and β coefficients were used to transform the risk model in the current risk score. The risk score was tested for external validation in a different population of patients in the present study. Considering the risk score, the predictor variables (collected at baseline) were history of Periodontitis (present or absent), bone level (located on the implants’ coronal third or implants’ middle third), lack of a passive fit or a non-optimal screw joint (presence or absence of prosthetic passive fit, prosthetic screw loosening, abutment screw loosening, prosthetic decementations), bacterial plaque (modified plaque index [17] and recorded as presence or absence), bleeding (modified bleeding index [17] and recorded as presence or absence), type of material used in the restoration (ceramic, metal–ceramic, metal–acrylic, acrylic resin), proximity of other implants/teeth (presence or absence) and current smoking status. Baseline measurements were taken one-year post-surgery. This time point was chosen to prevent overlap between early implant failures and peri-implant disease. A risk score was calculated from the sum of the scores for each predictor to estimate the likelihood of peri-implant disease, shown in Table 1. Risk groups were established using the total score for each patient and separating them into four categories of risk for peri-implant disease: low risk, moderate risk, high risk, and very high risk (Table 1).

The secondary outcome measure was the effect of the average time interval between maintenance appointments (≤6 months or >6 months) during follow-up on the incidence of peri-implant disease. This was evaluated for a subset of cases and matched controls using a nested case-control study design. Cases were defined as patients with peri-implant disease; controls were defined as patients without peri-implant disease during the 5-year follow-up. Sampling was performed by including all cases and randomly sampling controls matched for risk category for peri-implant disease (the high- and very high-risk groups were combined in order to sample from the controls). The average time interval between maintenance appointments was calculated by dividing the time between the surgery date and the diagnosis of peri-implant disease by the number of maintenance appointments in that period (for cases) or by dividing the total time of follow-up (60 months) by the number of maintenance appointments in that period (for controls). The average time interval between maintenance appointments was recoded into a dichotomous variable (≤6 months or >6 months).

### 2.4. Statistical Analyses

Descriptive statistics included means and standard deviations to describe quantitative variables and frequencies to describe qualitative variables. The prevalence of peri-implant disease was calculated. Inferential analysis was performed as follows. Relative risk (RR) for peri-implant disease for each “risk groups” was calculated using logistic regression. Receiver operating characteristic (ROC) curves were calculated along with their 95% confidence intervals (95% CI). The risk group discrimination was computed using c-statistics (95% CI). 

The effect that maintenance appointments with time intervals of >6 months had on the incidence of peri-implant disease was evaluated for a subset of cases and matched controls using conditional logistic regression for estimation of RR with 95% CI. The impact that 6-month or greater time intervals between maintenance appointments had on the incidence of peri-implant disease was estimated as the attributable risk percent (ARP) using the RR: ARP = (RR − 1) / RR [18]. The significance level was set at 5%. Statistics were performed using Statistical Package for Social Sciences (SPSS, version 17.0, IBM SPSS, New York, NY, USA).

## 3. Results

### 3.1. Primary Outcome Measure: Model Discrimination Capacity

The distribution of patients across risk groups was 102 patients with low risk (28.9%), 68 patients with moderate risk (19.3%), 77 patients with high risk (21.8%), and 106 patients with very high risk (30%) (Table 2). The incidence of peri-implant disease in the cohort was 24.1% (*n* = 85 patients). The distribution of peri-implant disease according to the groups of risk was two patients with moderate risk (2.9%), 22 patients with high risk (28.6%), and 61 patients with very high risk (57.5%).

“Risk group” was a significant risk factor in the logistic regression model with an RR = 5.52, (95% CI: 3.64, 8.35) and a high degree of significance (*p* < 0.001 for the goodness of fit) (Table 3). The model accuracy was 80.5%, with a sensitivity of 71.8% (61/85), and a specificity of 83.2% (223/268).

The area under the curve (95% confidence interval) was 0.858 (0.820, 0.896) (*p* < 0.001, c-statistic) (Figure 1). 

Risk scores are presented in Figure 2 and Figure 3. These scores can be used to obtain predictions for individual patients. As an example, we will examine a 59 year-old female with no history of periodontitis who was a non-smoker and who was rehabilitated with a three-unit implant-supported rehabilitation (three implants of 7 mm in length, positions #44–#46) that were restored using metal–ceramic material. At baseline (one year of follow-up) bacterial plaque and bleeding were present, and the bone level was located on the implants’ coronal third (Figure 2), which produces a total score of 8 points and puts her in the very high-risk profile. Peri-implant disease was diagnosed after 58 months on implant #45 due to the presence of peri-implant pockets of 5 mm and a three thread marginal bone loss (Figure 3).

### 3.2. Secondary Outcome Measure: Effect and Impact of Having > Six Months Intervals between Maintenance Visits 

The sub-sample included 85 cases (with peri-implant disease) and 85 controls (without peri-implant disease), matched by risk group. The distribution of the predictor variable (average interval between maintenance appointments) was as follows: 62 cases and 64 controls had intervals between maintenance appointments of ≤ six months (incidence in the non-exposed = 0.49) while 23 cases and 21 controls had intervals between maintenance appointments > six months (incidence in the exposed = 0.52). The conditional logistic regression model did not show a significant effect of the time interval between maintenance appointments when comparing the two group, with a RR = 1.13, (95% CI: 0.57, 2.21) (Table 4). The impact on the incidence of peri-implant disease assessed through the attributable risk percent was 5.89%.

## 4. Discussion

The current study validated a risk score obtained from a model in a different population of patients, with a high degree of significance (*p* < 0.001 for the goodness of fit) as well as an excellent discriminating ability (c-statistic = 0.858) for indicating which implant-supported rehabilitation patients had a greater risk for peri-implant disease during a five-year post-surgery follow-up period. This interpretation of the c-statistic result is supported by a previous publication that suggested that ROC curves falling on the diagonal will approach the following c-statistic values for discrimination capacity: 0.5 when there is no discriminating capacity to diagnose patients; 0.7–0.8 when there is acceptable discrimination; 0.8–0.9 when there is excellent discrimination capacity; and 0.9 or greater when there is outstanding discrimination [19]. This risk score thus represents a valid tool for aiding clinicians in evaluating a patient’s risk for peri-implant disease and providing important information at an early stage of follow-up with the objective of avoiding the condition. The prevalence of peri-implant disease has been reported across a broad range between 1% and 47% at the implant level, a variation that can be attributed to differences in the terminology, etiology, and diagnostic criteria of the condition [12]. The currently used terminology of “peri-implantitis” has been challenged previously but is a term for the use of unknown implant–host interfaces with a relatively good knowledge of the tooth–host interface (periodontitis) [20]. This condition was assumed for the current investigations [4,16]. The use of this definition is supported by recent genetic investigations that have revealed differences in gene expression between patients with peri-implant disease and those with periodontitis [21,22]. The etiology, which is considered to be a pathology largely produced by a bacterial infection of plaque accumulation [23], was previously refuted [4,16,23,24,25,26,27] after several factors aside from bacterial plaque emerged as potential risk factors in a multivariable analysis [16,23]. Aside from the risk model from which the present risk scores were derived [16], Konstantinidis et al. [25] also conducted a cross-sectional study of 168 patients with 597 implants to evaluate the prevalence of and possible risk indicators for peri-implant diseases. The study reported no association between high plaque scores and peri-implant disease. Other theories include bone loss that is a result of a combination of factors such as poorly manufactured implants, placement by untrained clinicians, and individual patient characteristics [27]. Alternatively, the condition could be a result of an immunologically derived rejection mechanism. In this theory, infection is considered to be a secondary event following a disturbance in the immune system due to a foreign-body, such as the dental implant. The foreign implant would then lead to immunological and inflammatory responses [28]. Additionally, different causal models have been proposed that do not include bacterial plaque as a component cause [4,26], such as the Sander Greenland component causal model [29]. This model states chronic diseases are generally multifactorial and thus more than one causal mechanism may induce the condition, with each causal mechanism being a product of the impacts of several component causes (risk factors). Moreover, the diagnostic criteria for the condition’s clinical manifestation also differ across studies, which makes it difficult to compare them [12]. Several risk factors included in the current validated risk score were retrieved from a risk model [16] and were previously highlighted as risk indicators in other studies using multivariable analysis. Not surprisingly, a history of periodontitis, bleeding, or bacterial plaque [25,30,31,32] are all biological risk indicators and are supported by a greater amount of research. However, other factors related to biomechanical aspects such as the presence of wear facets on the prosthetic crown were found to be significantly associated with peri-implant disease [32]. Gurgel et al. [31] conducted a cross-sectional study with 155 patients to evaluate the frequency of peri-implant diseases and factors associated with its occurrence. The study found that medication use, the existence of two or more implants, and a gingival bleeding index >10% were risk indicators for peri-implant disease when adjusted for other variables of interest. In a subsequent cross-sectional analysis, Konstatinidis et al. [25] evaluated the potential risk indicators for peri-implant disease using multi-level logistic regression models that registered both tooth loss due to periodontitis and implants in the maxilla as risk indicators. Dalago et al. [32] also conducted a cross-sectional study with 183 patients and 916 implants that aimed to identify systemic and local risk indicators associated with peri-implant disease. The study found significant associations between history of periodontal disease (odds ratio = 2.2), cemented restorations (odds ratio = 3.6), total rehabilitations (odds ratio = 16.1), and the presence of wear facets (odds ratio = 2.4). In contrast, Marrone et al. [30] found that in a study with 103 patients with 266 implants, there were no significant effects for peri-implant disease at the patient level using an age >65 years, total edentulism, hepatitis, active periodontitis, dental plaque ≥30%, Diabetes, history of Periodontitis, smoking, and dental visits ≤1 per year as explanatory variables. The authors attributed these results to having a lower statistical power relative to the high number of variables in the model. 

Previous authors have addressed the importance of refining treatment plans through the use of informed decision-making processes to improve the outcomes of implant-supported fixed prostheses. Controlling risk factors and anticipating outcomes are an essential part of implant dentistry risk management to prevent implant failure [33,34]. Moreover, controlling for risk would produce health gains for patients and reduce time spent in a dental chair to resolve any mechanical complications experienced [35].

The non-significant effect of the time interval between maintenance appointments on the incidence of peri-implant disease in the present study agree with the results of previous systematic reviews. Bidra et al. [14] conducted a systematic review to evaluate the evidence on patient recall and the maintenance of implant-supported restorations. The review concluded that there was minimal evidence related to recall regimens in patients with implant-supported restorations. This result inferred a “D” classification (the lowest classification) regarding the level of evidence and the qualitative reserve of not considering the recommendation as a standard of care [14]. Furthermore, Monje et al. [12] identified a significant effect of peri-implant maintenance therapy on the incidence of peri-implant disease, stressing the need to tailor the recall regimen to the patients’ risk profile and not to rely on the interval between maintenance appointments alone. The modest impact of the recall regimen registered in the present study (an increase of less than 6%) on the excess risk for peri-implant disease is corroborated by the finding of Monje et al. [12] and underlines the potential greater importance of each patients’ risk profile. A recent comment also highlighted this idea, stating that even when maintenance therapy is provided, biological complications can still occur and that maintenance therapy should be tailored to a patients’ risk profile [13]. For the avoidance of doubt, the result of the present study should not be interpreted as labeling the recall regimen irrelevant to the outcome. Rather, this study is suggesting that recall regimen is one piece of a more comprehensive approach to preventing peri-implant disease. A previous systematic review highlighted the paramount importance of individually tailored supportive programs that are based on patient motivation and re-instruction in oral hygiene measures and combined with professional implant cleaning to prevent peri-implant disease [11]. 

Given the implications that the validation of the risk score in the present study have for clinical practice, the authors propose a comprehensive approach to managing patient risk. This approach involves the measurement and management of, and communication about, risk. Measuring risk involves assessing the risk factors for peri-implant disease and using a risk score and risk profile for each patient. Managing risk is related to examining the implications of a patient’s risk profile result and the deciding the necessary action(s) that should be taken. This decision should consider the use of professional cleaning procedures, the maintenance instructions, patient motivation, the recall interval, or changes in the prosthetic complex. Communicating risk involves sharing information about the risk score with the patients, holding the patients responsible for their own health, and working together on short term goals (one recall schedule at a time) that can improve the results of the risk assessment (not only plaque control improvement but also smoking cessation and/or other lifestyle factors for example). Ultimately, this could improve the result of the risk profile and recall regimen, via a positive feedback mechanism. The rules necessary to apply the current risk score for prediction consist in patients rehabilitated with dental implants, with at least one year of follow-up and with the possibility of evaluating the patient’s periodontal history, clinical, and radiographic diagnosis. The risk score can be performed in any evaluation appointment from the one year of follow-up onwards, enabling the patient and clinician to work on the negative aspects present at that time. 

The main strength of this study is that to the authors’ best knowledge, it presents the first risk score for predicting peri-implant disease that has been validated for patients who were rehabilitated with implant-supported restorations. Moreover, the predictive risk score had excellent discriminating ability, which will enable precise predictions. The limitations of this study included the study conduction in a single center and one implant system (suggesting caution in extrapolating the results); the fact that all variables were collected at baseline and their effect accounted for prediction in the remaining of the follow-up irrespective of the status remaining unaltered or altering (for example: plaque control; a limitation in every prediction study); and the average time between maintenance appointments (as they could be unevenly distributed during the follow-up period) that was performed for standardization and analytical purposes. The multifactorial origin of peri-implant disease presents a challenge with direct implications in prediction across different populations as different variables may exert an effect on the risk of incidence, making it impossible to account in a single model and therefore posing a limitation of this study (such as different implant systems, implant micro and macro designs, prosthodontic options, or even clinician experience). Despite being validated in a different sample, studies of the different heterogeneous populations are needed to further strengthen the models’ external validity. 

## 5. Conclusions

Within the limitations of this retrospective study it can be concluded that a tool that enables the five-year prediction of peri-implant disease in patients with dental implants was validated. This tool could be used in implant dentistry to improve the prognosis for implant-supported restorations, providing a base for identifying patients at greater risk of peri-implant disease and enabling a stricter control. Maintenance appointments of < six months or > six months did not influence the incidence of peri-implant disease when considering the matching of cases and controls by risk profile.

## Figures and Tables

**Figure 1 jcm-08-00252-f001:**
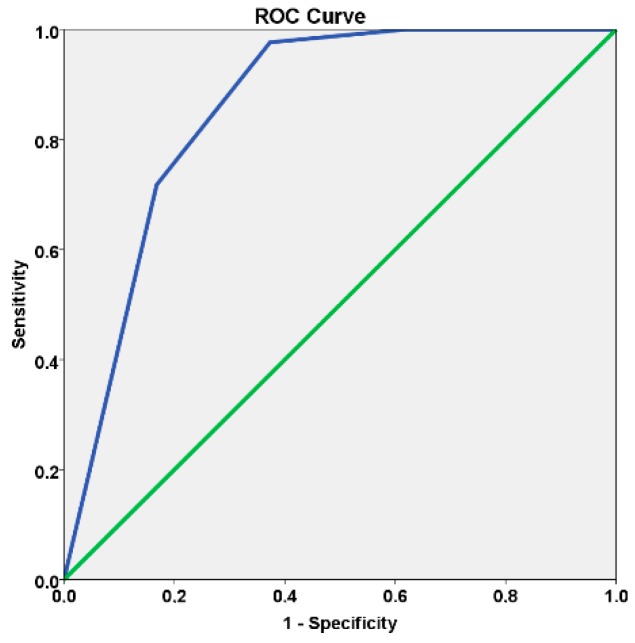
Receiver operating characteristic curve illustrating the performance of the risk algorithm (Area under the curve = 0.858, 95% confidence interval [0.820, 0.896], standard error = 0.019, *p* < 0.001, cut-off point = 0.14).

**Figure 2 jcm-08-00252-f002:**
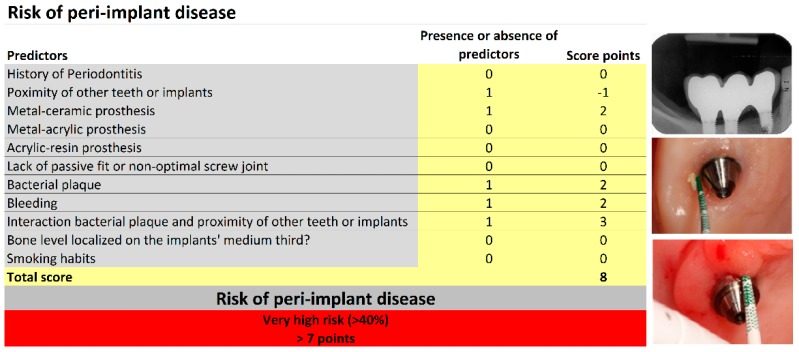
Baseline periapical radiograph (at one year of follow-up) from a 59 years old female patient with no history of periodontitis who was a non-smoker, rehabilitated with a three-unit implant-supported rehabilitation [three implants of 7 mm in length, positions #44–#46; Risk points = −1 due to adjacent implants or teeth present) restored using metal–ceramic material (Risk points = 2). Bacterial plaque (Risk points = 2) and bleeding (Risk point = 2) were present, and the bone level was located on the implants’ coronal third. Due to the presence of bacterial plaque and the presence of adjacent implants or teeth, 3 risk points were added. A total score of 8 points was registered for this patient which corresponded to the very high-risk group.

**Figure 3 jcm-08-00252-f003:**
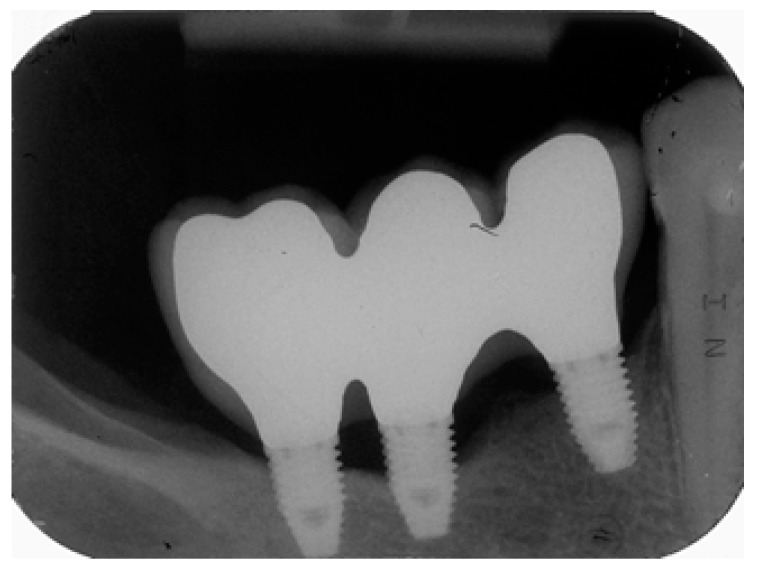
Periapical radiograph at 58 months of follow-up with peri-implant disease diagnosed in implant #45 with the presence of peri-implant pockets of 5 mm and a three threads marginal bone loss.

**Table 1 jcm-08-00252-t001:** Baseline patient characteristics, risk score points and illustration of the relationship between the points system and the risk profile groups.

Variable			Total (*n* = 353) *n* (%)	Risk Score Points *
History of Periodontitis			137 (38.8)	5
Bone level located on the implants’ middle third			20 (5.7)	5
Lack of prosthetic fit or non-optimal screw joint (passive misfit abutment screw loosening, prosthetic screw loosening, prosthetic decementations)			20 (5.7)	3
Bacterial plaque			187 (53)	2
Bleeding			175 (49.6)	2
Type of material used in the restoration				
Ceramic			79 (22.4)	0
Metal–ceramic			186 (52.7)	2
Metal–acrylic			104 (29.5)	−2
Acrylic			42 (11.9)	1
Adjacent implants or teeth present			214 (60.6)	−1
Simultaneous presence of bacterial plaque and adjacent implants or teeth present			90 (25.5)	3
Smoker			83 (23.5)	0
Risk profile groups obtained from the sum of risk score points for each variable				
Risk profile	Low risk	Moderate risk	High risk	Very high risk
Number of points	<2 points	2–4 points	5–7 points	>7 points

* De Araújo Nobre et al. 2016 [4].

**Table 2 jcm-08-00252-t002:** Distribution of demographics and predictors according to the risk groups.

Variables	Low Risk Total of 102 Patients	Moderate Risk Total of 68 Patients	High Risk Total of 77 Patients	Very High Risk Total of 108 Patients
**Demographics**	*N* (within group %)	*N* (within group %)	*N* (within group %)	*N* (within group %)
**Average age in years**	49.5	52.7	53.5	54.6
**Sex**				
**Female**	56 (54.9%)	50 (73.5%)	45 (58.4%)	60 (56.6%)
**Male**	46 (45.1%)	18 (26.5%)	32 (41.6%)	46 (43.4%)
**Predictors**	N (within group %)	N (within group %)	N (within group %)	N (within group %)
**History of Periodontitis**	0 (0%)	13 (19.1%)	43 (55.8%)	81 (76.4%)
**Bone level located on the implants’ middle third**	0 (0%)	0 (0%)	2 (2.6%)	18 (17%)
**Lack of prosthetic fit or non-optimal screw joint**	0 (0%)	2 (2.9%)	2 (2.6%)	16 (15.1%)
**Bacterial plaque**	6 (5.9%)	37 (54.4%)	49 (63.6%)	95 (89.6%)
**Bleeding**	11 (10.8%)	35 (51.5%)	42 (54.5%)	87 (82.1%)
**Type of restoration material**				
**Ceramic**	39 (38.2%)	13 (19.1%)	15 (19.5%)	12 (11.3%)
**Metal–ceramic**	45 (44.1%)	26 (38.2%)	39 (50.7%)	76 (71.7%)
**Metal–acrylic**	22 (21.6%)	35 (51.5%)	29 (37.7%)	18 (17%)
**Acrylic**	6 (5.9%)	4 (5.9%)	9 (11.7%)	23 (21.7%)
**Adjacent implants/teeth present**	84 (82.4%)	26 (38.2%)	41 (53.2%)	63 (59.4%)
**Interaction presence of bacterial plaque * adjacent implants/teeth present**	0 (0%)	9 (13.2%)	25 (32.5%)	56 (52.8%)
**Smoker**	19 (18.6%)	12 (17.6%)	22 (28.6%)	30 (28.3%)

**Table 3 jcm-08-00252-t003:** Logistic regression model for the association between “risk group” and the incidence of peri-implant disease.

Variable	B	Standard Error	Significance (*p*-Value)	Relative Risk (RR)	95% Confidence Interval for RR	
					Lower	Upper
Risk Group	1.71	0.212	*p* < 0.001	5.52	3.64	8.35
Constant	−6.41	0.753	*p* < 0.001	0.002		

R-square = 0.46; Omnibus test: *p* < 0.001; Hosmer and Lemeshow test: *p* = 0.139). Model evaluation parameters: sensitivity, 71.8%; specificity, 83.2%; accuracy, 80.5%.

**Table 4 jcm-08-00252-t004:** Conditional logistic regression model for the association between the average time interval between maintenance appointments and the incidence of peri-implant disease.

Variable	B	Standard Error	Significance (*p*-Value)	Relative Risk (RR)	95% Confidence Interval for RR	
					Lower	Upper
>6 months between maintenance appointments	0.118	0.344	*p* = 0.732	1.13	0.57	2.21

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
