# Peer review of "A Peri-Implant Disease Risk Score for Patients with Dental Implants: Validation and the Influence of the Interval between Maintenance Appointments"

_jcm, 2019, doi:10.3390/jcm8020252_

Reviewer 1 Report

Dear authors,

I have read your work very carefully and I want to congratulate you for the effort you have made. This study is based on the evaluation and monitoring of a very large sample of patients and its clinical implication is of great impact for the management of patients with implants.

However, I would like to indicate some aspects that are recommended to improve for publication:

- The data regarding the prevalence of peri-implant disease are incorrect. The confidence intervals are changed. I found in the original article that the data would be as follows: Weighted mean implant-based and subject-based peri-implantitis prevalences were 9.25% (95% Confidence Interval (CI): [7.57, 10.93]) and 19.83% ( CI [15.38, 24.27) respectively. They must modify it. (line 33).

- In the calculation of the regression it is desirable to incorporate the adjusted coefficient of determination to assess the percentage explained by the variance taking into account the inclusion of all the variables. (Table 2). An r2 of 0.46 is an unacceptable coefficient to explain the model. Can the authors explain anything more about this interpretation and its implication for prediction?

- For eminently clinical readers, could you detail the rules necessary to apply your prediction ?.

Thanks for your effort.

Author Response

Dear authors,

I have read your work very carefully and I want to congratulate you for the effort you have made. This study is based on the evaluation and monitoring of a very large sample of patients and its clinical implication is of great impact for the management of patients with implants.

However, I would like to indicate some aspects that are recommended to improve for publication:

1- The data regarding the prevalence of peri-implant disease are incorrect. The confidence intervals are changed. I found in the original article that the data would be as follows: Weighted mean implant-based and subject-based peri-implantitis prevalences were 9.25% (95% Confidence Interval (CI): [7.57, 10.93]) and 19.83% ( CI [15.38, 24.27) respectively. They must modify it. (line 33).

Response: The authors thank the Reviewer’s indication. The figures were changed as indicated.

Changes: Page 1, lines 33 and 34

2- In the calculation of the regression it is desirable to incorporate the adjusted coefficient of determination to assess the percentage explained by the variance taking into account the inclusion of all the variables. (Table 2). An r2 of 0.46 is an unacceptable coefficient to explain the model. Can the authors explain anything more about this interpretation and its implication for prediction?

Response: The authors thank the Reviewer’s query and respect his opinion. However, the r2 of 0.46 being unacceptable coefficient to explain the model is subject of discussion. First, Moksony 1990 considers that the r2 statistic to be highly misused as the most important indicator of the quality of a study, referring that it is obviously important in prediction to know how accurately the dependent variable can be estimated from the explanatory variables, however, the coefficient of determination actually is a poor measure of how close the estimated values come to the observed ones. Second, it really depends on what the investigation is trying to model, this means that it is dependent on the context: If what was trying to be predicted was molecule behavior, then an r2 of 0.46 would be too low, as the prediction in molecule research can be performed farely well; however, when modeling anything related to human behavior or potential intervention, a value of r2 =0.46 is excellent. If one investigates the general values for the r2 in the research in modeling chronical diseases or outcomes in Medicine it is observed reports for the r2 of 0.17 for diabetes (Agarwal et al. 2016), 0.168 for associations between medical student empathy and personality (Costa et al. 2014) or 0.268 for using 3D bone texture analysis as a potential predictor of radiation-induced insufficiency fractures (Nardone et al. 2018); common to the 3 manuscripts, the consideration that the model was discriminative with an area under  the curve statistic of 0.74 for the models of Costa et al. 2014 and Nardone et al. 2018.

This is further supported by a study from Cohen 1992 that reported r2 values of 0.02, 0.13 and 0.26 as small, medium and large r2 values, respectively. But more important than the R2 could be the measure of how close the estimated values come to the observed ones, and this was evaluated in our study (and the references cited in this response) through the area under the curve statistics with 0.858 which represents excellent discrimination capacity. The authors did include in the Discussion section a discussion of the multifactorial origin of peri-implant disease and its implication in prediction across different populations with different characteristics (including implant macro and micro designs).

References:

-Moksony F. Small is beautiful. The use and interpretation of R2 in social research. SzociolĂłgiai Szemle, Special issue:130-138.

-Costa P, Alves R, Neto I, MarvĂŁo P, Portela M, Costa MJ. Associations between medical student
empathy and personality: a multi-institutional study. PLoS One 2014; 9:e89254. doi: 10.1371/journal.pone.0089254. eCollection 2014.

-Nardone V, Tini P, Croci S, Carbone SF, Sebaste L, Carfagno T, et al. 3D bone texture analysis as a potential predictor of radiation-induced insufficiency fractures. Quant Imaging Med Surg 2018;8:14-24. doi: 10.21037/qims.2018.02.01.

-Cohen J. A power primer. Psychological Bulletin 1992; 112; 1:155-159.

Changes: Discussion section, lines 346-350

3- For eminently clinical readers, could you detail the rules necessary to apply your prediction ?. Thanks for your effort.

Response: The authors thank the Reviewer’s suggestion. The rules necessary to apply the prediction consist in patients rehabilitated with dental implants, with at least one year of follow-up and with the possibility of evaluating the patient’s Periodontal history, clinical and radiographic diagnosis. The risk score can be performed in any evaluation appointment from the one year of follow-up onwards, enabling the patient and clinician to work on the negative aspects present at that time.

Changes: Discussion section, lines 335-340

Reviewer 2 Report

The manuscript belongs to the scope of the journal, is well written and provides a potential resource with extensive application in the dental field (risk score). Also, involves the evaluation of medical and local risk factors  that may influence the incidence of the disease.

However, there are some clarifications required before publication.

Therefore this reviewer recommends minor review.

COMMENTS TO THE ABSTRACT

No comments

COMMENTS TO THE INTRODUCTION

-The definition for peri-implant disease used by the authors is too old (1984). This reviewer recommends to use the most recent definitions for the disease Please use the following references and re-define peri-implant disease. 

Renvert S, Persson G, Pirih F, Camargo P. Peri-implant health, peri-implant mucositis, and peri-implantitis: Case definitions and diagnostic considerations. J Periodontol. 2018 Jun;89 Suppl 1:S304-S312. doi: 10.1002/JPER.17-0588.

Schwarz F, Derks J, Monje A, Wang HL. Peri-implantitis. J Periodontol. 2018 Jun;89 Suppl 1:S267-S290.

Berglundh T, Armitage G, Araujo MG, Avila-Ortiz G, Blanco J, Camargo PM, Chen S, Cochran D, Derks J, Figuero E, Hämmerle CHF, Heitz-Mayfield LJA, Huynh-Ba G, Iacono V, Koo KT, Lambert F, McCauley L, Quirynen M, Renvert S, Salvi GE, Schwarz F, Tarnow D, Tomasi C, Wang HL, Zitzmann N. Peri-implant diseases and conditions: Consensus report of workgroup 4 of the 2017 World Workshop on the Classification of Periodontal and Peri-Implant Diseases and Conditions.

 J Periodontol. 2018 Jun;89 Suppl 1:S313-S318

COMMENTS TO MATERIALS AND METHODS

-Page 2, Lines 70-71. Please correct  the following sentence " The inclusion criterion was patients rehabilitated and followed by our team for 5 years after their surgery" and replace by " Patients with rehabilitated dental implants with at least 5 years of follow-up after the  insertion of the rehabilitation"

-Page 2, Lines 72 to 75. Please clarify the following paragraph " Exclusion criteria were patients not followed by our team and 72 patients with implants that failed or had unresolved biological complications during the first year of follow-up (early failures) whose signs and symptoms could overlap with the clinical manifestations of peri-implant disease."

Please define which were the "unresolved biological  complications"

Also as per this reviewer, early implant failure is such implant that fails before the prosthetic restoration is inserted. Please provide the reference for "implant early failure" that you used.

-Page 2, Lines 88 and 89. Please clarify the following sentence " The final sample included a total of 353 patients (211 female, 142 male) with an average age 88 (standard deviation) of 52.5 years (11.75 years)" 

What is (11.75 Years)? Might be a misspelling error?

-Page 3, Lines 101 to 107. The authors mentioned  the predictor variables : history of periodontitis, bone level, lack of passive fit or non-optimal screw joint, bacterial plaque, bleeding, type of material, proximity of other implants or teeth and current smoking status.

Why the authors did not included as a predictive variable,  the type of prosthetic restoration (cemented, screw retained, hybrid restoration?

This is a very important factor that can influence clinical practice, this reviewer recommends to include this variable in addition to the variables of Mombelli et al, 1987, that were used by the authors.

-Page 4, Lines 114 to 125. In relation to the following paragraph " The secondary outcome measure was the effect of the average time interval between maintenance appointments (≤ 6 months or > 6 months) during follow-up on the incidence of peri-implant disease. This was evaluated for a subset of cases and matched controls using a nested case-116 control study design"

There is a serious concern of this reviewer.  The incidence just measure episodes of peri-implant disease but not  their characteristics, duration and severity  and patient related factors among others.

This could underestimate  the real impact of the maintenance appointments on reduction or control of peri-implant diseases and not just their incidence.  Therefore, the validity of your statement at the conclusions "The impact of a longer recall regimen was negligible"  might be compromised.

This reviewer recommends to include your data related to prevalence of the peri-implant disease  on your patient's pool for a better understanding.

Also the authors should develop more  in the discussion section this issue.

COMMENTS TO DISCUSSION

See previous comments

COMMENTS TO CONCLUSIONS

-Page 8, Lines 297 and 298. Please correct the following sentence "The effect of a time interval between maintenance  appointments of > 6 months on peri-implant disease was modest" and replace by "Maintenance  appointments of<6 months="" or="">6 months did not influence the incidence of peri-implant disease"

In addition, please start your conclusions with the following words " within the limitations of this retrospective study can be concluded"

Author Response

The manuscript belongs to the scope of the journal, is well written and provides a potential resource with extensive application in the dental field (risk score). Also, involves the evaluation of medical and local risk factors that may influence the incidence of the disease.

However, there are some clarifications required before publication.

Therefore this reviewer recommends minor review.

COMMENTS TO THE ABSTRACT

1-No comments

Response: Thank you. Considering the changes performed in the response to points 7, 8 and 9 of this review, the Abstract was also amended.

Changes: Abstract section, lines 27 and 28.

COMMENTS TO THE INTRODUCTION

 2-The definition for peri-implant disease used by the authors is too old (1984). This reviewer recommends to use the most recent definitions for the disease Please use the following references and re-define peri-implant disease.

Renvert S, Persson G, Pirih F, Camargo P. Peri-implant health, peri-implant mucositis, and peri-implantitis: Case definitions and diagnostic considerations. J Periodontol. 2018 Jun;89 Suppl 1:S304-S312. doi: 10.1002/JPER.17-0588.

Schwarz F, Derks J, Monje A, Wang HL. Peri-implantitis. J Periodontol. 2018 Jun;89 Suppl 1:S267-S290.

Berglundh T, Armitage G, Araujo MG, Avila-Ortiz G, Blanco J, Camargo PM, Chen S, Cochran D, Derks J, Figuero E, Hämmerle CHF, Heitz-Mayfield LJA, Huynh-Ba G, Iacono V, Koo KT, Lambert F, McCauley L, Quirynen M, Renvert S, Salvi GE, Schwarz F, Tarnow D, Tomasi C, Wang HL, Zitzmann N. Peri-implant diseases and conditions: Consensus report of workgroup 4 of the 2017 World Workshop on the Classification of Periodontal and Peri-Implant Diseases and Conditions.  J Periodontol. 2018 Jun;89 Suppl 1:S313-S318

Response: The authors thank the Reviewer’s suggestions. Some of the references the authors do not agree with them as they have severe limitations:

First, Berghlund et al. 2018, a consensus (with all the disadvantage associated with the context of a consensus), despite defining the condition in a similar way as Albrektsson 1984, considers it an exclusive plaque-associated condition, a situation that is clearly not the case as demonstrated in the present investigations; second, Renvert et al. 2018, while applying the same exclusive plaque (or biofilm) induced pathology, considers the absence of previous radiographs (this is paramount) and a radiographic bone level >/= 3 mm in combination with bop and peri-implant pockets >/=6mm as peri-implantitis, discarding the possibility that for example,  implants inserted in grafted bone may present decreased bone levels, or implants with 10+ years of follow-up that may present a lower marginal bone level that in the presence of a mucositis could produce the same signs. Therefore the authors included the definition of disease that is more in agreement with our investigations (Schwarz et al. 2018). However, for the authors it is important to keep the first reference as this work is based on it and it started before the publication of any of the references suggested.

Changes: Introduction, lines 32-34 and References (reference 2) sections.

COMMENTS TO MATERIALS AND METHODS

 3-Page 2, Lines 70-71. Please correct  the following sentence " The inclusion criterion was patients rehabilitated and followed by our team for 5 years after their surgery" and replace by " Patients with rehabilitated dental implants with at least 5 years of follow-up after the  insertion of the rehabilitation"

Response: The authors thank the Reviewer’s indications. The text was amended as suggested.

Changes: Materials and Methods section, lines 74-76   

4-Page 2, Lines 72 to 75. Please clarify the following paragraph " Exclusion criteria were patients not followed by our team and 72 patients with implants that failed or had unresolved biological complications during the first year of follow-up (early failures) whose signs and symptoms could overlap with the clinical manifestations of peri-implant disease."

Please define which were the "unresolved biological  complications"

Also as per this reviewer, early implant failure is such implant that fails before the prosthetic restoration is inserted. Please provide the reference for "implant early failure" that you used.

Response: The authors thank the Reviewer’s queries. Unresolved biological complications were complications developed in the first year of follow-up such as abscess, infection or fistulae that were not able to be resolved through non-surgical or surgical interventions during that first year (during osseointegration) and therefore overlapping in clinical signs with peri-implant disease. The authors clarified the manuscript. Considering that peri-implant disease is a condition occurring in osseointegrated dental implants the cut-off point in follow-up was 1 year. The same for the early implant failure whose definition was considered as the failure during osseointegration. Nevertheless, it is dependent on the loading regimen: As the Reviewer correctly declared, early implant failure is such implant that fails before the prosthetic restoration is inserted (this is applied to 2-stage surgical rehabilitations as per Brånemark protocol or 1-stage surgical rehabilitations with implant insertion and abutment connection); however, when preforming immediate loading, early failure is such implant that fails during the osseointegration period but after the prosthetic restoration as the restoration is performed on the day of surgery (although provisional). The reference for early implant failure was inserted as suggested for avoidance of doubt.

Reference:

Jemt T, Olsson M, Franke Stenport V. Incidence of First Implant Failure: A Retroprospective Study of 27 Years of Implant Operations at One Specialist Clinic. Clin Implant Dent Relat Res. 2015;17:e501-10. doi: 10.1111/cid.12277. Epub 2014 Dec 23.

Changes: Materials and Methods section, lines 77-79, References section (reference 15) 

5-Page 2, Lines 88 and 89. Please clarify the following sentence " The final sample included a total of 353 patients (211 female, 142 male) with an average age 88 (standard deviation) of 52.5 years (11.75 years)" 

What is (11.75 Years)? Might be a misspelling error?

Response: The authors thank the Reviewer’s query. This was the standard deviation. For avoidance of doubt, we moved the definition right next to the value.

Changes: Materials and Methods, line 94.

6-Page 3, Lines 101 to 107. The authors mentioned  the predictor variables : history of periodontitis, bone level, lack of passive fit or non-optimal screw joint, bacterial plaque, bleeding, type of material, proximity of other implants or teeth and current smoking status.

Why the authors did not included as a predictive variable, the type of prosthetic restoration (cemented, screw retained, hybrid restoration?

This is a very important factor that can influence clinical practice, this reviewer recommends to include this variable in addition to the variables of Mombelli et al, 1987, that were used by the authors.

Response: The authors thank the Reviewer’s query. The authors did include a significant number of variables during the construction of the risk score that was published in a previous publication (de Araújo Nobre 2015). The variables were retained in the final model considering their significance and biological plausibility and from that model a risk score was constructed. The current manuscript is the validation of the risk score in a different sample from where the model was derived so to test external validation and that was the reason why the current risk score was tested this way, because the evaluation of potential risk indicators was already performed in a previous publication. Nevertheless the authors introduced the variables tested in the construction of the model for the avoidance of doubt and so to make this manuscript more independent. The particular case of type of prosthetic restoration was dealt with in the variable lack of prosthetic fit or non-optimal screw joint (through the evaluation of decementations) and no hybrid restorations were present.

Changes: Materials and Methods section, lines 106-143

7-Page 4, Lines 114 to 125. In relation to the following paragraph " The secondary outcome measure was the effect of the average time interval between maintenance appointments (≤ 6 months or > 6 months) during follow-up on the incidence of peri-implant disease. This was evaluated for a subset of cases and matched controls using a nested case-116 control study design"

There is a serious concern of this reviewer.  The incidence just measure episodes of peri-implant disease but not their characteristics, duration and severity and patient related factors among others.

This could underestimate the real impact of the maintenance appointments on reduction or control of peri-implant diseases and not just their incidence.  Therefore, the validity of your statement at the conclusions "The impact of a longer recall regimen was negligible"  might be compromised.

This reviewer recommends to include your data related to prevalence of the peri-implant disease  on your patient's pool for a better understanding.

Also the authors should develop more  in the discussion section this issue.

Response: The authors thank the Reviewer’s queries. The nested case-control study design was “nested” from the same sample of 353 patients during the same follow-up of 5 years with 85 cases (peri-implant disease patients) and 85 controls (peri-implant healthy patients) at the end of the follow-up matched by risk group (every case matched with a control belonging to the same risk group) as per protocol of standard case-control study designs. The incidence is the outcome of interest for the evaluation of the impact of maintenance appointments (if the disease occurred or not) and not the characteristics, duration and severity of disease (as for these to be measured the disease needs to be present in a first place). Patient related factors were accounted for precisely when the cases and controls were matched by risk groups considering the risk score. The exercise of this particular investigation was to test the impact of the maintenance appointments considering the risk profile of the patients and the result suggests that the interval between maintenance appointments was not particularly strong (~6%). This does not mean that maintenance appointments are not important but rather that they need to be tailored to the risk profile of each patient, a situation that was already present in the Discussion section of this manuscript. The authors acknowledge that the term “negligible” might be exaggerated and therefore replaced the term and clarified the manuscript to avoid misinterpretation.   

Changes: Abstract section, lines 26-28; Results section, lines 229-230; Discussion section, line 317; Conclusion section, lines 359, 360.

COMMENTS TO DISCUSSION

8-See previous comments

Response: The authors thank the Reviewer’s indication. The text was amended as suggested following the previous comments

Changes: According to previous comments.

COMMENTS TO CONCLUSIONS

9-Page 8, Lines 297 and 298. Please correct the following sentence "The effect of a time interval between maintenance  appointments of > 6 months on peri-implant disease was modest" and replace by "Maintenance  appointments of<6 months="" or="">6 months did not influence the incidence of peri-implant disease"

In addition, please start your conclusions with the following words " within the limitations of this retrospective study can be concluded"

Response: The authors thank the Reviewer’s indications. The text was amended as requested.

Changes: Conclusion section, lines 355, 356, 359, 360

Reviewer 3 Report

Authors study seems to be based on older classification of periimplantitis, whereas in the meantime newer classification is available.

How many patients came from private rehabilitation center and the Faculty in medicine? Authors stated later that fixed prosthesis and implants were installed only in private practice (P.2, line 70, P. 2, line 78). Were patients from University of Lisbon actually in the study at all?

Authors examined several important risk factors, but why such important parameters like implant surface, surgeon experience, type of restoration cement were not included in the classification?

P.3, line 97. Classification of peri-implant disease is confusing. For example, peri-implantitis is defined as presence of pocket => 5 mm and attachment loss => 2 mm compared to previous evaluation. What about the patients, which fulfil only one of these criteria?

P.3, line 107. Term “proximity” should be defined more in details.

Table 1. The calculation of risk factors raises several critical issues. First, the presence of bacterial plaque is considered two times. Second, The presence of bacterial plaque and lack of prosthetic fit are factors, which can controlled during the therapy. Third, what does it mean “bone level located on the implants’ middle third”?

P.4, line 121-125. The average time interval between appointments is not very informative parameter, because these appointments can be unevenly distributed during treatment period.

Inclusion and exclusion criteria must be specified more in detailed. What were the lowest and the highest age of patients? Patients with systemic diseases included/excluded? Smoking? Etc.

Demographic characteristics, smoking status, smoking load etc should be provided for patients belonging to different risk groups.

More details on statistical analysis must be provided. How logistic regression model was made? Which variables were considered as dependent, independent, covariates?

ROC curve (Figure 1) is used for binary classification system. How it is considered by Authors? What is the cut-off level for this curve?

Figure 2. Better radiographic picture showing the full length of implant must be presented.

The authors have already published the study on risk score for periimplantitis patients (De Araújo Nombre et al. J. Prosthodont Res 2017). It seems that the present manuscript use the same patients’ collective. The difference between both studies must be clearly stated.

Author Response

1-Authors study seems to be based on older classification of periimplantitis, whereas in the meantime newer classification is available.

Response: The authors thank the Reviewer’s suggestions. Some of the references suggested by another Reviewer the authors do not agree with them as they have severe limitations:

First, Berghlund et al. 2018, a consensus (with all the disadvantage associated with the context of a consensus), despite defining the condition in a similar way as Albrektsson 1984, considers it an exclusive plaque-associated condition, a situation that is clearly not the case as demonstrated in the present investigations; second, Renvert et al. 2018, while applying the same exclusive plaque (or biofilm) induced pathology, considers the absence of previous radiographs (this is paramount) and a radiographic bone level >/= 3 mm in combination with bop and peri-implant pockets >/=6mm as peri-implantitis, discarding the possibility that for example,  implants inserted in grafted bone may present decreased bone levels, or implants with 10+ years of follow-up that may present a lower marginal bone level that in the presence of a mucositis could produce the same signs. Therefore the authors included the definition of disease that is more in agreement with our investigations (Schwarz et al. 2018). However, for the authors it is important to keep the first reference as this work is based on it and it started before the publication of any of the references suggested.

Changes: Introduction, lines 32-34 and References (reference 2) sections.

2-How many patients came from private rehabilitation center and the Faculty in medicine? Authors stated later that fixed prosthesis and implants were installed only in private practice (P.2, line 70, P. 2, line 78). Were patients from University of Lisbon actually in the study at all?

Response: The authors thank the Reviewer’s query. The full sample of patients was rehabilitated and followed at the private practice, while data analysis and interpretation were performed at the University of Lisbon. The manuscript was amended for clarity.

Changes: Materials and Methods section, line 69.

3-Authors examined several important risk factors, but why such important parameters like implant surface, surgeon experience, type of restoration cement were not included in the classification?

Response: The authors thank the Reviewer’s query. The authors did include a significant number of variables during the construction of the risk score that was published in a previous publication (de Araújo Nobre 2015). The variables were retained in the final model considering their significance and biological plausibility and from that model a risk score was constructed. The current manuscript is the validation of the risk score in a different sample from where the model was derived so to test external validation and that was the reason why the current risk score was tested this way, because the evaluation of potential risk indicators was already performed in a previous publication. Nevertheless the authors introduced the variables tested in the construction of the model for the avoidance of doubt and so to make this manuscript more independent.

Changes: Materials and Methods section, lines 106-143

4-P.3, line 97. Classification of peri-implant disease is confusing. For example, peri-implantitis is defined as presence of pocket => 5 mm and attachment loss => 2 mm compared to previous evaluation. What about the patients, which fulfil only one of these criteria?

Response: The authors thank the Reviewer’s query. The clinical signs for peri-implant disease were not clear due to a mistyping: peri-implant pockets ≥ 5 mm, bleeding upon probing, the concurrent presence of vertical bone loss visible in the periapical radiograph compared to the previous evaluation and/or attachment loss ≥ 2 mm. This because to calssifiy as peri-implant disease the clinical signs in an incident case are increased pocket depth and bleeding on probing (but these alone could be also mucositis) together with concurrent presence of vertical bone loss (this is mandatory for peri-implant disease) and/or clinical attachment loss (which is a clinical manifestation of vertical bone loss). The text was adapted.

Changes: Materials and Methods section, line 103

5-P.3, line 107. Term “proximity” should be defined more in details.

Response: The proximity means right next to the implant using the international notation(implant in position #14, tooth in position #13)

Changes: Materials and Methods section, lines 116,117.

6-Table 1. The calculation of risk factors raises several critical issues. First, the presence of bacterial plaque is considered two times. Second, The presence of bacterial plaque and lack of prosthetic fit are factors, which can controlled during the therapy. Third, what does it mean “bone level located on the implants’ middle third”?

Response: The authors thank the Reviewer’s query. Plaque was considered once, the other presence is the interaction between plaque and proximity (this is validated in logistic regression-Katz 2006); plaque and lack of prosthetic fit are factors which can be controlled during therapy but for plaque, the majority of patients did not (despite being one of the limitations in prognostic research with logistic regression-the authors include it in the Discussion section). Concerning lack of prosthetic fit, this variable was not tested alone, it was part of a broader definition (Lack of prosthetic fit or non-optimal screw joint) that besides passive misfit, also included abutment screw loosening, prosthetic screw loosening, prosthetic decementations) that are events that occur. The focus was not if the variable could be controlled or not during therapy (obviously in case of passive misfit the same was corrected), but rather if it was present and what effect could induce in the incidence of peri-implant disease. Bone level located on the implants’ middle third consisted in patients with implants that at one year of follow-up had the bone level on the implants’ middle third: these represented patients for example with implants inserted in grafted bone with a higher bone loss during osseointegration, implants that were inserted and had the incidence of an infection that deemed a bone level on the implants’ middle third after non-surgical or surgical interventions, or implants inserted with a dehiscence), all these cases represented a low incidence rate of 5.7%.

Reference: Katz, M.H. Multivariable analysis: A practical guide for clinicians. 2nd ed.; Katz, M.H., Eds.; University Press, Cambridge, United Kingdom, 2006.

Changes: Discussion section, lines 346-349

7-P.4, line 121-125. The average time interval between appointments is not very informative parameter, because these appointments can be unevenly distributed during treatment period.

Response: The authors thank the Reviewer’s indication. It is true but it was necessary to perform a standardization of the maintenance recall periods during the 5 years of follow-up. The authors included it as a limitation in the Discussion section.

Changes: Discussion section, lines 349-351

8-Inclusion and exclusion criteria must be specified more in detailed. What were the lowest and the highest age of patients? Patients with systemic diseases included/excluded? Smoking? Etc.

Response: The authors thank the Reviewers’ queries. The age range was reported as requested (18 to 87 years) and the inclusion criteria adapted to clarify that the patients were included in the study irrespective of having a systemic condition or being smokers.

Changes: Materials and Methods section, lines 74-76 and 94.

9-Demographic characteristics, smoking status, smoking load etc should be provided for patients belonging to different risk groups.

Response: The authors thank the Reviewer’s indication. The information requested was inserted in a new table (Table 2).

Changes: Results section, Table 2.

10-More details on statistical analysis must be provided. How logistic regression model was made? Which variables were considered as dependent, independent, covariates?

Response: The authors thank the Reviewer’s queries. The authors included a significant number of variables during the construction of the risk score that was published in a previous publication (de Araújo Nobre et al. 2015). The variables were retained in the final model considering their significance and biological plausibility and from that model a risk score was constructed. The current manuscript is the validation of the risk score in a different sample from where the model was derived so to test external validation, hence the reason why the current risk score was tested this way, as variable testing was already performed in a previous publication. Nevertheless the authors introduced the variables tested in the construction of the model for the avoidance of doubt and so to make this manuscript more independent.

Changes: Materials and Methods section, lines 106-143

11-ROC curve (Figure 1) is used for binary classification system. How it is considered by Authors? What is the cut-off level for this curve?

Response: The authros thank the Reviwer’s query. The ROC curve is used as a discrimination between patients with and without the disease combining the values of true positive and true negative cases in order to assess the level of prediction. The cut-off level for the curve was 0.14 and was inserted for clarity.

Changes: Figure 1, line 204.

12-Figure 2. Better radiographic picture showing the full length of implant must be presented.

Response: The authors tank the Reviewer’s indication. However in the image (which is purely illustrative) it is clear that the bone level is stable around the implants and therefore it is the authors’ opinion that it serves the purpose; Moreover, as there is no other periapical radiograph at baseline for this particular case it would need to be provided a completely new case for illustration and therefore the authors kindly ask the Reviewer to leave Figure 2 as it stands.

Changes: None.

13-The authors have already published the study on risk score for periimplantitis patients (De Araújo Nombre et al. J. Prosthodont Res 2017). It seems that the present manuscript use the same patients’ collective. The difference between both studies must be clearly stated.

Response: The authors thank the Reviewers’ indication. The study on the derivation of the risk score was performed and published previously as the Reviewer indicated while the present study focuses on the validation of the risk score in a different sample of patients in order to test the external validity of the risk score. This is a common practice in prediction research. The differences between both studies and the scientific justification were inserted in the manuscript for clarity as suggested.

References: 

-Grendar J, Jutric Z, Leal JN, Ball CG, Bertens K, Dixon E, Hammill CW, Kastenberg Z, Newell PH, Rocha F, Visser B, Wolf RF, Hansen PD. Validation of Fistula Risk Score calculator in diverse North American HPB practices. HPB (Oxford) 2017;19:508-514. doi: 10.1016/j.hpb.2017.01.021. Epub 2017 Feb 21.

-Tham T, Costantino P. Comparison of venous thromboembolism risk stratification models in a high risk otolaryngology patient cohort. J Perioper Pract 2019 23:1750458919826794. doi: 10.1177/1750458919826794.

Changes: Materials and Methods section, lines 141-143; Discussion section, lines 240,241

Round  2

Reviewer 3 Report

Authors answered all queries.